# p53 Promotes Cytokine Expression in Melanoma to Regulate Drug Resistance and Migration

**DOI:** 10.3390/cells11030405

**Published:** 2022-01-25

**Authors:** Pinakin Pandya, Lyubov Kublo, Jacob Stewart-Ornstein

**Affiliations:** Department of Computational and System Biology, Hillman Cancer Center & University of Pittsburgh, Pittsburgh, PA 15232, USA; LKUBLO@pitt.edu

**Keywords:** wild type p53, melanoma, NF-kB, cytokines, BRAF inhibitor

## Abstract

The transcription factor p53 is frequently lost during tumor development in solid tumors; however, most melanomas retain a wild type p53 protein. The presence of wild type p53 in melanoma has fueled speculation that p53 may play a neutral or pro-tumorigenic role in this disease. Here we show that p53 is functional in human melanoma cell lines, and that loss of p53 results in a general reduction in basal NF-kB regulated cytokine production. The reduced cytokine expression triggered by p53 loss is broad and includes key inflammatory chemokines, such as CXCL1, CXCL8, and the IL6 class cytokine LIF, resulting in a reduced ability to induce chemotactic-dependent migration of tumor cells and immune cells and increased sensitivity to BRAF inhibition. Taken together, this result indicates that wild type p53 regulates cytokine expression and induces cytokine-dependent phenotype on melanoma.

## 1. Introduction 

The pleiotropic transcription factor p53 is a tumor suppressor that is frequently lost in human cancer development [1]. Some types of cancers, however, show relatively low p53 mutation rates. In melanoma, despite an extremely high average mutation burden, p53 is often retained, with only ~20% of melanoma showing a mutation in p53 [2,3]. Further, these p53 mutations are not associated with reduced survival [4,5], and melanoma is relatively rare in families with germline p53 mutations [6,7]. These features of melanoma have fueled speculation that p53 may play a complex and potentially pro-tumor role in this cancer through the regulation of cell cycle arrest [8]. 

Previously, we observed that p53 could associate with cytokine genes such as IL1A and CXCL1 in certain melanoma cell lines [9], and other groups have shown an association between p53 status and cytokine secretion in fibroblasts [10] and macrophages [11]. This suggested to us that in melanoma p53 could potentially regulate—directly or in partnership with other factors—the expression of key immune-regulatory cytokines. As many cytokines regulate the local immune environment (e.g., pro-inflammatory cytokines) and promote proliferation through paracrine and autocrine signaling, this p53-regulated secretion could provide p53 wild type (p53^WT^) melanoma cells with an advantage relative to their p53 loss-of-function (p53^lof^) counterparts. We therefore set out to test if loss of p53 would reduce the ability of melanoma cells to secrete cytokines and resist therapy. 

In cancer cells, p53 can be inactivated by direct mutation or loss of the p53 gene or indirectly through perturbation of the p53 network [12,13]. We show that in four melanoma cell lines, treatment with DNA-damaging agents resulted in increased abundance of p53 and expression of classical downstream target genes. Loss of p53 in these cell lines resulted in reduced expression of p53 target genes, but also loss of NF-kB-regulated cytokines such as CXCL1. Indeed, multiplex cytokine measurements showed loss of several critical pro-inflammatory and pro-growth secreted cytokines including CXCL1, VEGF, LIF, and IL1. This loss of cytokine expression resulted in reduced ability to attract migrating cells and reduced intrinsic cellular fitness, including vulnerability to BRAF inhibitors.

## 2. Materials and Methods 

### 2.1. Cell Culture and Cell Treatment 

The UACC62, UACC257, LOX-IMVI, and SK-MEL-5 lines were grown in RPMI media (Gibco or VWR) + 10% FBS (Hyclone) + 1% PFS (Gibco) at 5% CO2 at 37 °C. All cell lines were obtained from ATCC, and with the exception of the hTERT-immortalized melanocytes, all have been previously verified by string profiling. The hTERT-immortalized melanocytes were obtained from ATCC and were grown in the human melanocyte cell culture system (GIBCO—medium 254 supplemented with human melanocyte growth supplement) at 5% CO2 at 37 °C. 

Cells were treated with Nutlin-3A (5 µM, Sigma from a stock concentration of 10 mM in DMSO), etoposide (25 µM, Sigma, stock of 50 mM in DMSO) as indicated. For UV treatment cells were switched from normal medium to DPBS (Gibco) and exposed to UVC (254 nm, 10 J/m^2^) using a mounted UV lamp; the media was subsequently replaced with fresh warm media, and the cells returned to the incubator for the indicated period of time.

For the stimulation experiment, melanoma cells were treated with 0.1 µg/mL recombinant Wnt5a (r-Wnt5a, R and D from a stock concentration of 10 µg/mL in phosphate-buffered saline containing 0.1% bovine serum albumin) for 16 h in serum-free medium or incubated with 10 ng/mL TNF-alpha (TNF-α, from stock concentration of 50 µg/mL) for the indicated time in complete medium. 

To collect supernatants, p53 wild type (p53^WT^) and knockout (p53^ko^) melanoma cells were seeded in 35 mm dishes (1 × 10^5^ cells/dish) or in 100 mm culture dishes (8 × 10^5^ cells/dish) in complete media and cultured for 48 h at 37 °C and 5% CO2-incubator. Culture media was harvested, centrifuged at 1000× *g* for 10 min (4 °C), filtered and stored at −20 °C. For Transwell migration assay, medium was replaced with serum-free medium, and culture dishes were kept in incubator for 24 h before harvesting supernatants. 

To induce p53 expression in knockout cell lines, we infect UACC62 cells with p53-GFP cloned into a FUGW-puro lentiviral vector (FU-H2B-GFP-IRES-Puro was a gift from Charles Gersbach (Addgene plasmid # 69550; http://n2t.net/addgene:69550; RRID:Addgene_69550)) (accessed on 18 April 2019). The p53 sequence was cloned in place of the H2B sequence after digestion with XBA1 and AGE1. 

### 2.2. Reagent and Antibodies 

Nutlin-3A was from Sigma-Aldrich(St. Louis, MO, USA). Recombinant Wnt5a and human TNF-α was obtained from R & D system (Minneapolis, MN, USA)and Peprotech (Cranbury, NJ, USA) respectively. Mouse monoclonal antibodies for p53 protein (DO-1), β-actin (MA1-140) and rabbit monoclonal Abs for phospho-ERK1/ERK2 (Thr185, Tyr187) (700012) were from Thermo Fisher Scientific Technology (Waltham, MA, USA). Rabbit monoclonal antibodies for NF-kB p65 (8242) were from Cell Signaling Technology Inc. (Danvers, MA, USA). Rabbit polyclonal antibodies against N-terminal PARP1 (39559) was from Active Motif (Carlsbad, CA, USA). APC-conjugated mouse monoclonal anti-NF-kB p65 was obtained from BioLegend (San Diego, CA, USA). Infrared (IR)-conjugated goat anti-mouse (IRDye 680) and goat anti-rabbit (IRDye 800) were from LI-COR Bioscience (Lincoln, NE, USA). NucBlue^TM^ live cell staining reagent was from Invitrogen Bioscience (Waltham, MA, USA). Puromycin (10 mg/mL), blasticidin (10 mg/mL) and G418 antibiotic solution were from Biovision (Milpitas, CA, USA) and VWR life science (Bridgeport, NJ, USA), respectively. 

### 2.3. Lentiviral Production and Infection 

The human embryonic kidney cells (HEK-293Tx) were seeded into a 60mm culture dish (5 × 10^5^ cells/dish). At 70–80% cell confluency, constructs were co-transfected into HEK-293Tx cells together with three plasmids encoding Gag, Pol, Rev and VSVG (third generation virus production system) using transIT-LT1 transfection reagent (Mirus, Madison). Next day, the reagent containing medium was replaced with fresh complete medium. After 24 h incubation, viral supernatant was collected, centrifuged at 800× *g* for 10 min, filtered through a 0.45 µm filter, and stored at −80 °C or used immediately. Melanoma cells, which were plated on the previous day and reached a 50–60% confluency level, were infected by the addition of viral supernatant. 24 h after infection, the media was removed, cells were washed once and supplied with complete medium containing selection agents. 

### 2.4. Generation of p53^ko^ Lines 

LOX-IMVI and SK-MEL-5 cell lines were infected overnight with Cas9-blast (gifts from John Doench & Willian Hahn & David Root (Addgene plasmid #96924) virus. Then, cells were washed once with PBS-1X and supplied with fresh complete culture media containing blasticidin (10 µg/mL, Biovision). The selection process was continued for one week by delivering fresh medium with antibiotic every 2 or 3 days. Cells expressing Cas9 were verified by Western blot, and frozen down as a poly-clonal population. 

Melanoma cells expressing Cas9 were infected with guide RNA expressing virus (lentiGuide-Puro was a gift from Feng Zhang (Addgene plasmid # 52963; http://n2t.net/addgene:52963; RRID:Addgene_52963) (accessed on 14 October 2019), guide sequence ATCTGACTGCGGCTCCTCCA) that is targeting the n-terminus of the endogenous p53 sequence or non-targeting sequence (GTATTACTGATATTGGTGGG) as a control. Next day, cells were washed once with PBS-1x, the media was replaced by puromycin (1.5 µg/mL, Invitrogen) containing complete RPMI media, and the selection process continued for 3 days. Antibiotic-resistant cells were isolated and cloned through limiting dilution in 96-well plates. After the clonal expansion step, the p53 protein expression was determined by Western blot analysis, and individual clones with no p53 protein expression were considered as knockout clones. 

UACC62 and UACC257 cells were transfected with cas9-guide RNA plasmid with a guide targeting p53 (pX330-U6-Chimeric_BB-CBh-hSpCas9, ATCTGACTGCGGCTCCTCCA) using transIT-LT1 transfection reagent. Transfected cells were treated with Nutlin-3A (5 µm) for 7 days, followed by single-cell cloning by limiting dilution to obtain p53 knockout clones. p53^ko^ cell lines were confirmed by immunoblot analysis. 

### 2.5. Preparation of Cellular Lysates and Western Blotting 

Cells were resuspended in a lysis buffer (50 mM tris pH 7.5, 100 mM NaCl, 1% Triton X-100, 0.5% sodium deoxycholate, 0.1% sodium dodecyl sulfate combined with protease and phosphatase inhibitor cocktail [Halt^®^ protease and phosphatase cocktail, Thermo]) followed by 30 min incubation on ice and centrifugation at 13,000 rpm for 30 min at 4 °C. Cleared supernatant was collected and stored at −80 °C. The collected supernatant was quantitated using the Bio-Rad protein assay reagent (Bio-Rad, Hercules, CA, USA). 

Protein lysates (equivalent to 15 to 25 µg protein) were combined with loading dye and reducing agent (DTT), followed by vortexing and kept at 95 °C for 5 min. The boiled protein samples were resolved on 4–12% Bis-tris gradient gel (Invitrogen) and transferred onto 0.4 µM nitrocellulose membrane. The membrane was blocked using 3% BSA in 0.1% tween−20 containing tris-buffered saline, and incubated with primary antibodies overnight at 4 °C. After washing three times with TBST-1x (5 min for each washing step), membrane was incubated with IRDye conjugated goat anti-mouse or anti-rabbit secondary antibodies for 1h in the dark at RT, and immunoreactive protein bands were detected using a typhoon imaging system (Amersham, GE, USA). 

The primary antibodies used were as follows: anti-p53 (1:1000), anti-actin (1:1000), anti-pERK1/pERK2 (1:1000), anti-p65 (1:1000) and anti-PARP1 (1:1000) in blocking solution (3% BSA in 0.1% tween−20 containing TBS-1X). 

### 2.6. RNA Extraction and qPCR 

Melanoma cells were seeded in a 35 mm dish and cultured at 37 °C and 5% CO2 incubator. They were collected in Trizol at a 70–80% confluency point. Total RNA was extracted using the direct zol RNA kit (Zymo Research Corp, Irvine, CA, USA) according to the manufacturer’s instructions. qPCR was performed on a Quantstudio3 (Thermo). Total RNA (typically 100 ng) was reverse transcribed using the LunaScript RT super mix (NEB, Ipswich, MA, USA) and cDNA was diluted 10-fold. Each PCR reaction was run as technical duplicates or triplicates using 5µL of Luna Universal qPCR master mix (NEB), 1 µL of each 2.5 µM primer, and 3 µL of diluted cDNA. The PCR program was run according to the manufacturer’s suggestions and quantification was done using the quantstudio software. Subsequent analysis of qPCR ct values was performed in Prism (Graphpad), normalizing to ACTB expression.

Primers used are as follows

ACTB: ACCTTCTACAATGAGCTGCG, CCTGGATAGCAACGTACATGG

IL1A: TGTATGTGACTGCCCAAGATG, TTAGTGCCGTGAGTTTCCC

Il1B: ATGCACCTGTACGATCACTG, ACAAAGGACATGGAGAACACC

CXCL1: AACCGAAGTCATAGCCACAC, CCTCCCTTCTGGTCAGTTG

CXCL8: AGGGTTGCCAGATGCAATAC, AAACCAAGGCACAGTGGAAC [14]

LIF: CCAGATCAGGAGCCAACT, CCAAGGTACACGACTATGC [15]

IL6: CAGTTCCTGCAGAAAAAGGCAA, AGCTGCGCAGAATGAGATGA [16]

WNT5A: ATTCTTGGTGGTCGCTAGGTA, CGCCTTCTCCGATGTACTGC [17]

TNFAIP3: CTGCCCAGGAATGCTACAGATAC, GTGGAACAGCTCGGATTTCAG [18]

P65: AATGGCTCGTCTGTAGTGC, TGCTCAATGATCTCCACATAGG

NF-kB1: GAAAAGCTGTAAACATGAGCCG, ACCCTGACCTTGCCTATTTG

### 2.7. TCGA Analysis 

Data from the Pan-Cancer Atlas [19] were downloaded from cBioPortal [20,21]. The following datasets were used from the TCGA Pan-Cancer Atlas [22]: “skcm_tcga_pan_can_atlas_2018”. The “data_RNA_Seq_v2_expression_median” was extracted to measure gene expression and the “data_mutations_extended” was used to identify tumors with a p53 mutation, defined as the presence of any mutation in p53 that was not classified as ‘silent’ or ‘3′UTR’. Tumor IDs between RNA and DNA data were then matched, and the gene expression profiles were sorted into p53 wild type or p53 mutant. From this data, the expression of each gene in the dataset was compared between the p53 mutant and wild type groups of tumors with a KStest. Genes were then sorted by their significance and sign of effect (e.g., from most to least significant for up- and down-regulated genes) and genes with less than an average FPKM (fragment per kilobase of transcript per million base read) of 2 were discarded. Gene set enrichment analysis was obtained from the GSEA2 Gene Set Enrichment Analysis (GSEA) function in MATLAB [23]. 

### 2.8. mRNA Sequencing and Analysis 

Total RNA was extracted from cells using trizol (thermo) and DirectZol miniprep kit (Zymo). RNA (500 ng) was used to generate libraries using the Ultra II mRNA seq kit with poly-a selection (NEB), following the manufacturer’s protocols. Libraries were quantified with qPCR (NEB library quant kit) and sequenced on a Nextseq500 with 1 × 75 bp reads. Samples were obtained and libraries generated in biological duplicates. Reads were aligned to the Hg19 refseq transcripts using Salmon V0.7.2 [24], and reads were aggregated from transcript to genes. Gene read tables were subsequently compared using Deseq2 [25]. Heatmaps and gene intersections were generated with MATLAB or R.

Gene set enrichment analysis was obtained from the GSEA2 Gene Set Enrichment Analysis (GSEA) function in MATLAB [26]. Genes were sorted by fold change in expression after applying cutoffs for minimal expression (>25 read counts). Where groups of genes were obtained rather than ranked lists (e.g., overlapping gene sets across multiple cell types [Figure 1] or multiple knockouts [Figure 2]), a simple hypergeometric test was used to obtain enrichment for different Hallmark gene sets using the Broad/UCSC web server [26,27,28]. 

### 2.9. Si-RNA Mediated Gene Silencing 

Gene silencing using a mixture of 4 different siRNAs targeting Wnt5a (L-003939-00-0005), p65 (L-003533-00-00005) or non-targeting control (D-001810-01-05) were received from Dharmacon Inc. (ThermoFisher Scientific, Waltham, MA, USA). Transfection steps proceeded as per manufacturer instruction. Briefly, cells (5 × 10^4^ cells/well) were seeded in 12-well plates and cultured for overnight. Adherent cells were transfected using mixture of si-RNA (5 nm) and dharmafect transfection reagent (2.5 µL/well) in reduced serum media (Opti-MEM). 48 h post-siRNA delivery, cells were collected in trizol and utilized for gene expression analysis by qRT-PCR. 

### 2.10. Transwell Migration Assay 

Transwell migration assay was performed in a Boyden chamber containing polycarbonate filters with 8 µm pore size (costar, Bodenheim, Germany). LOX-IMVI cells were harvested by trypsin for 2 min and mixed with defined trypsin inhibitor (cascade biologics, ThermoFisher Scientific) in a ratio of 1:1. The Transwell upper compartment was loaded with 100 µL cellular suspension (6 × 10^5^ cells/mL in serum-free media), whereas the lower compartment was filled with supernatant (600 µL) that was collected from p53^WT^ or p53^ko^ melanoma cells or serum-free culture medium for negative control. After incubation at 37 °C and 5% CO2-incubator for 24 h, Transwell filters were removed, fixed using 4% PFA for 15 min at RT, and stained by NucBlue Live stain reagent (Invitrogen) for 1 h. Transwell images were photographed by LionHeart/FX automated microscope (BioTek, Santa Clara, CA, USA), and total migrated cells in each well were counted by FiJi image J software. 

For U-937 cell migration assay, cells (1 × 10^6^/mL) were suspended in serum-free medium, and 100 µL cellular suspension was loaded into the upper compartment. The lower compartment was filled with 600 µL p53^WT^ or p53^ko^ supernatant or serum-free culture medium. After incubation at 37 °C and 5% CO2-incubator for 24 h, Transwell filters were removed and cells in lower compartment were counted using Cellometer (Nexcelom Bioscience, Lawrence, MA, USA).

### 2.11. Cytokine Expression Analysis 

Tumor cells supernatants were collected as mentioned before. Human CXCL1 expression was analyzed using Elisa set (R & D system, DY 275), according to company instruction. The culture supernatants were assayed at a dilution within the linear range of the CXCL1 standards, and the concentration of CXCL1 in each sample was determined using a standard curve, as indicated by the kit. 

Multiplex cytokines analysis was performed using tumor cell supernatants by human 71-cytokine /chemokine array panel kit (Milipore corporation, Burlington, MA, USA). Experimental and analytical steps were carried out at Eve technology corporation (Calgary, AB, Canada).

### 2.12. Immunostaining and Microscopy 

Cells (1 × 10^4^/well) were seeded in 96-well black glass bottom plates (Cellvis, Mountain View, CA, USA) and left untreated or treated with TNF-α (10 ng/mL, 15 min) at a 70–80% confluency point. Cells were fixed with 2% PFA for 10 min, permeabilized and blocked with 2% BSA in PBS-1X consisting of 0.1% triton X-100 for 1 h at room temperature. APC-conjugated NF-kB (14G10A21 Biovision, Milpitas, CA, USA), 1:400 diluted in blocking buffer) was incubated at 4 °C in humidified chamber overnight. Next day, they were washed twice with PBS-1x and subsequently incubated with NucBlue Live stain reagent for 1h in the dark. Pictures were taken with an inverted Nikon TI2 microscope (Melville, NY, USA), LED light source ( Lumencor, Parkway Beaverton, OR, USA), and CMOS camera (prime 95B, Photometrics, Tucson, AZ, USA), and analyzed using Fiji imageJ software. 

### 2.13. Cell-Survival Analysis 

Cell survival analysis was performed using cell titer-Glo^®^ 2.0 assay (Promega, Fittsburgh, MA, USA) according to instructions provided by company. Briefly, cells (3 × 10^3^/well) were seeded in 96-well microtiter plates and kept at 37 °C in a 5% CO2-incubator. Next day, cells were treated with serial dilution of vemurafenib (16 µm to 1 µm) or DMSO (control) for 48 h in incubator. Then, cells were exposed to cell titer-Glo^®^ 2.0 reagent for 10min at RT, and the obtained luminescence value was measured by plate reader (BioTek Synergy/LX multimode reader). The relative cell survival analysis was calculated in relation to DMSO treated samples.

### 2.14. Statistical Analysis of qPCR, Western Blots, Cell Survival and Cytokine Expression 

Statistical analysis was performed using GraphPad Prism 8.0.0 software (GraphPad software, Inc.). For comparative mRNA expression analysis of p53^WT^ vs. p53^ko^ by qPCR analysis, studies were conducted by biological quadruplicates and technical duplicate (if varies in biological samples, then it is mentioned in the figure legend). Cytokine analysis experiment was conducted in biological triplicate and technical duplicates. Western blot analysis was carried out in two independent experiments. Cell survival analysis was conducted in three independent experiments, and each experiment was carried out using technical quintuplicate (*n* = 5) wells, technical replicates were averaged, and biological replicates were used to compute statistics. Statistical differences between groups were determined using unpaired Student’s *t*-test or ANOVA (one-way or two-way). *p* < 0.05 was considered a statistically significant difference. 

## 3. Results

### 3.1. Melanoma Cell Lines Retain a Functional p53

One hypothesis of why melanoma cells frequently retain wild type p53 is that they inactivated the p53 pathway by alternative mechanisms. To test this, we treated four common melanoma cell lines (LOX-IMVI, SK-MEL-5, UACC62 and UACC257) with DNA-damaging agents (UV and etoposide) or the non-genotoxic MDM2 inhibitor and p53 activator Nutlin-3A [29]. We observed in each cell line that p53 levels rose three hours after treatment for the etoposide and Nutlin-3A conditions and weakly for the UV treatment (Figure 1A). Though p53 levels rise in response to treatment, it is possible that p53 signaling in these models is defective for the transcriptional activation of key cell cycle arrest or pro-apoptosis target genes. 

To more generally explore if p53 in melanoma cells can regulate a common set of p53 target genes, we used mRNA seq to monitor gene expression in each of the four cell lines after p53 activation by UV, etoposide, or Nutlin-3A. We identified those genes that were upregulated (>1.5-fold change, pval < 0.01) in all four cell lines in at least one condition (*n* = 85, Figure 1B). Among these target genes, many are canonical p53 target genes, such as p21 (CDKN1A), MDM2, and NOXA (PMAIP1), including both pro-apoptotic and cell cycle arrest factors. Overall, the p53 pathway (Hallmark gene set) was strongly enriched in this set of overlapping upregulated genes (Figure 1C). Interestingly, our analysis also suggested that the NF-kB/TNF pathway was also induced by these treatments, as were apoptotic and epithelial-to-mesenchymal (EMT) signaling (Figure 1C). To visualize the extent to which these pathways are activated in each cell line, we generated gene set enrichment analysis (GSEA) plots for each treatment (Figure 1D etoposide treatment, and Appendix A for Nutlin-3a and UV treatments) for the p53 and NF-kB pathways. We observed strong enrichment for p53 signaling in each cell line, and enrichment of NF-kB/TNF signaling in each line in the majority of conditions (the LOX-IMVI line is not significant for the Nutlin-3A treatment, and the SK-Mel5 line is not significant for Nutlin-3a and UV treatment; Figure 1D, Appendix A). We note that given the overlap between the NF-kB and p53 signatures (of the 200 genes in each set 26 are shared), disentangling the two pathways can be challenging informatically. We used qPCR to verify the activation of p53 target genes in each cell line and observed robust activation of p21 in all conditions and MDM2 in etoposide and Nutlin-3A treatment consistent with our RNAseq data (Figure 1E). 

To test if these transcriptional effects are observed in normal cells as well as cancer models, we treated hTERT-immortalized melanocyte cells with the MDM2 inhibitor Nutlin-3a. Using mRNAseq we observed strong activation of p53 signaling and also NF-kB targets (Appendix A) in these non-cancerous melanocytes, suggesting conservation of this pathway beyond cancer.

To determine if p53 activity in vitro is related to human cancer models, we used mRNAseq gene expression data from the Pan-Cancer Atlas (TCGA) to compare human melanoma samples with wild type p53 to those with mutations in p53 [19,22]. We observed that compared to wild type cancers, p53 mutant melanomas show greatly reduced p53 signaling (Figure 1F) and reduced mRNA expression of the key p53 target gene p21 (CDKN1A, Figure 1G). These results argue that, as in cell culture in most melanomas which retain wild type p53, this protein/gene expression network is active. 

### 3.2. Loss of p53 in Melanoma Cell Lines Results in Reduced p53 and NF-kB Signaling 

To directly test the role of p53 in melanoma cells, we used Cas9 to knockout p53 from the LOX-IMVI and SK-MEL-5 cell lines and derived clones from single cells (Figure 2A). We note a weak lower molecular weight band in one LOX-IMVI clone that could be a truncation product (clone #2), however, our data suggest this clone is also a functional knockout. Using mRNAseq we compared three different p53^ko^ clones for their gene expression relative to wild type. Knockout of p53 produced reproducible changes in gene expression across the three clones, with 387 and 147 genes changing significantly (>0.5 log2-fold change, *p* value < 0.01) in all clones from the LOX-IMVI and SK-MEL-5, respectively.

Enrichment analysis of the overlapping genes altered in the p53 knockout cells (Figure 2B) showed strong enrichment for p53 signaling and also NF-kB signaling (especially in the LOX-IMVI cells). Combined with our earlier results showing changes in NF-kB signaling in conditions that activate p53 (Figure 1), these results suggest that in the LOX-IMVI and SK-MEL-5 cell lines’ loss of p53 results in a substantial change in NF-kB signaling. 

To extend this analysis and identify commonalities between the two cell lines, we identified overlapping genes altered on p53 deletion. The genes altered in LOX-IMVI and SK-MEL-5 showed only modest, albeit significant, overlap (*p* = 1.09 × 10^−4^; hypergeometric test, applied to test the significance of overlap between two gene sets). Analyzing the 39 genes that were reduced, and one gene that was upregulated in at least 2/3 clones in both cell lines we find that this overlap is enriched in both p53- and NF-kB-regulated genes. We also note the presence of certain genes (e.g., FN1—fibronectin) that are involved in EMT signaling (Figure 2C,D).

These results suggested that p53 loss might act on NF-kB signaling to induce non-canonical phenotypes. In melanoma, some of the key targets of NF-kB are immune modulatory cytokines. We therefore explored in a more targeted way using qPCR if we observed changes to key cytokines on p53 deletion in melanoma cell lines (Figure 2E). We used qPCR to look at the canonical p53 target gene p21, and a set of cytokines (CXCL1, LIF, CXCL8, IL1A, IL1B, WNT5A) in three p53 knockout clones of LOX-IMVI, SK-MEL-5, UACC62, and UACC257 (Figure 2A and Appendix A). As we expected from our sequencing data, loss of p53 in each cell line resulted in reduced p21 expression, and also reduced expression of one or more of a heterogeneous set of cytokines (Figure 2E and Appendix A). We note that reduced CXCL1 expression is universal across the four cell lines (except clone1 in LOX-IMVI cells, where the trend is down but non-significant). These results are consistent with a model whereby p53 loss reduces NF-kB signaling, with complex effects on individual NF-kB target genes depending on the cell line context.

To further test the dependence of these effects on p53, we re-expressed p53 in the UACC62 p53^ko^ cell line (Appendix A) and observed a strong increase in p21 expression as expected, and a more modest increase in CXCL1, CXCL8, and LIF expression (Figure 2E). These results suggest that p53 regulates the expression of key cytokine genes in melanoma cell lines.

### 3.3. p53 Loss Reduces Secretion of Cytokines 

To determine if the transcriptional loss of cytokine expression also occurs at the protein level, we employed a cytokine array that measured the levels of 71 cytokines in supernatant harvested from equivalent numbers of p53^WT^ or p53^ko^ LOX-IMVI cells. Fifty-two out of seventy-one ligands were observed in detection range in p53^WT^ LOX-IMVI cells. These ligands are arranged in decreasing concentration range, i.e., >10, 1–10, and <10 pg/mL (Figure 3A). Testing three different knockout cell lines, we observed significant reductions of many key cytokines that mirrored our observations in mRNAseq (Figure 3A).

For example, CXCL1, IL1, LIF, and VEGFA were all significantly reduced (between ~2-fold for CXCL1, and >10-fold for LIF and VEGFA, Figure 3B). To validate this reduced cytokine expression, we ran an Elisa for CXCL1 on supernatant from the LOX-IMVI and SK-MEL-5 cell lines and observed in both models p53^ko^ clones showed reduced expression of CXCL1 (Figure 3C), consistent with our array and RNA measurements (Figure 2C and Figure 3A).

### 3.4. Key Cytokines Reduced by p53 Loss Are Regulated by NF-kB

Our sequencing analysis pointed to a potential role for NF-kB downstream of p53. Using the LOX-IMVI cell line as our model, we tested the regulation of cytokines such as CXCL1 by NF-kB. Using siRNA against the NF-kB (RelA) we observed a loss of NF-kB protein (Figure 4A), and reduced expression of CXCL1 and IL1, confirming the regulation of these targets by NF-kB in this system (Figure 4B).

If p53 loss results in a hard block to NF-kB expression, then we would expect activation of NF-kB by TNF would be attenuated. We therefore treated the p53^WT^ and p53^ko^ LOX-IMVI cell lines with TNF-α to activate NF-kB signaling. We observed that the p53^ko^ and p53^WT^ cells showed a comparable increase in gene expression for the canonical NF-kB target TNFAIP3 (Figure 4C). For cytokines such as CXCL1 where we observed reduced expression in p53^ko^ cells, treatment with TNF-α induced these target genes to similar (CXCL1, IL1B) or somewhat lower (IL6, WNT5A) levels compared to wild type (Figure 4D–G). We also observed relatively little change in the abundance and localization of the p65 protein in the p53 knockout cell lines Figure 4H and Appendix A). This suggests that loss of p53 reduces basal NF-kB signaling, without altering its ability to be activated by external signals.

Previously it was suggested that p53 regulated or was regulated by Wnt5a, which drove many of the p53 associated phenotypes in melanoma [8]. Though we do observe loss of Wnt5a expression in 2/4 of our cell lines, including the LOX-IMVI line (Figure 2E), knockdown of Wnt5a does not reduce CXCL1 expression (Appendix A). Further, the addition of recombinant Wnt5a does not result in increased cytokine expression in the LOX-IMVI p53^ko^ cell lines (Appendix A). These results suggest that Wnt5a is not playing a major role in the phenotypes we observe.

### 3.5. Loss of p53 Results in Reduced Cell Migration and Sensitivity to BRAF Inhibitor 

Cytokines are critical for driving cell migration and specifically CXCL1 has previously been shown to regulate cell movement [30]. We used a transwell assay to determine if loss of p53-driven cytokines affected cell migration. Supernatant from p53^WT^ LOX-IMVI cells could attract cancer cells through the transwell (Figure 5A,B), whereas supernatant from p53^ko^ cell lines showed reduced ability to drive cells through a transwell that was statistically indistinguishable from media. These results show that tumor cell migration is reduced in the context of the p53^ko^ secretome.

To test if the change in these results holds for immune cell migration, we repeated the experiment with the human monocyte line U937. As with the cancer cells, we observed that monocyte migration in a transwell was strongly promoted by wild type, but not p53^ko^ LOX-IMVI conditions media (Figure 5C).

Given that autocrine and paracrine signaling can be critical to driving resistance to certain therapeutic agents, we hypothesized that p53 knockout cells might be more sensitive to common therapeutic modalities. Indeed, previous work showed that the ability of p53 to block the cell cycle could promote survival in melanoma cells treated with DNA damaging therapies [8]. We therefore sought to test if loss of p53 would increase the sensitivity of LOX-IMVI cells to BRAF inhibitors which are commonly used to treat melanomas. Despite having a BRAF^V600E^ allele, LOX-IMVI cells are highly resistant to the BRAF inhibitor vemurafenib [31,32]. We observed that p53^WT^ LOX-IMVI cells show a little reduction of ERK phosphorylation when treated with vemurafenib (BRAF inhibitor) for 24 h; however, the p53^ko^ lines show reductions in ERK phosphorylation when treated with the same dose (Figure 6A). Consistent with reduced ERK activity, we observed that the p53^ko^ clones are more sensitive towards vemurafenib treatment as measured by cell titer-Glo (Figure 6B). We also noted a marked morphological change in the p53^ko^ cell lines upon treatment with vemurafenib that is absent in the wild type cells (Appendix A).

To explore the nature of this sensitivity more fully, we compared p53^WT^ and p53^ko^ cells at 1, 3, 6, and 24 h after vemurafenib treatment (2 µM). We observed that both genotypes have reduced pERK expression after 1 h however, pERK expression level rebounds in p53^WT^ cells to nearly basal levels at 24 h, while it remains suppressed in the p53^ko^ lines (Figure 6C). In these conditions we also observe the presence of cleaved PARP1 in the p53^ko^ cell line, which is an early marker of apoptosis [33,34]. 

## 4. Discussion 

Melanoma is unusual among solid tumors for its low rate of p53 mutations, which is especially surprising given its high UV-induced mutation rate. Other groups have suggested that p53 may provide some benefit to a developing melanoma by regulating the cell cycle through Wnt signaling [8]. Here we used unbiased profiling to identify cytokine regulation as significantly altered in multiple melanoma cell lines where we induce p53 loss. 

In the present study, using four different human melanoma cell-lines that possess wild type p53, we found the p53 is functionally active in these lines by demonstrating p53 activation and induce expression of p53 target gene expression in response to DNA-damage agents (UV or etoposide) or by Nutlin-3A, a specific p53 activator [29]. Further, our gene expression data suggested a potential role for p53 in the regulation of NF-kB target genes, consistent with the earlier observation from our group that used ChIPseq to show p53 DNA binding at CXCL1 in the LOX-IMVI cell lines. To decipher the role of p53 in melanoma, we performed transcriptome analysis in control and p53 knockout melanoma cells. In addition to a loss of canonical p53 transcription, we also identified a general reduction in many NF-kB target genes including cytokines at both the RNA and protein level. A linkage between p53 and cytokine expression is also observed in fibroblast [10] and macrophage cells [11]. 

Pro-inflammatory cytokines/chemokines are strongly regulated throughout melanoma development. For example, overexpression of CXCL1 is involved in primary cancer development and also during metastasis by promoting angiogenesis [35]. Additionally, work with mouse models of melanoma has also shown that inflammatory signaling is a strong promoter of metastasis in UV-induced melanoma [36]. 

Though we observed dampened expression of NF-kB target genes in p53-deficient melanoma cells, TNF-α treatment activates NF-kB and increases the expression of its target genes in melanoma cell lines regardless of p53 status. These results suggest that p53 regulates the basal activity of NF-kB but is not intrinsically required for its activation. Previous studies have shown that these two transcription factors interact to execute their functions. For example, NF-kB regulated expression of NOXA and p53AIP1 is required for p53-mediated cellular apoptosis in some cases [37,38]. Additionally, p53 is an indispensable component for stimulation of classical and atypical NF-kB target genes during replication stress [39]. In human macrophages, these transcription factors cooperate to induce pro-inflammatory cytokines and chemokines, including CXCL1, IL-6, and CXCL8 [11]. However, the exact dependency between NF-kB and p53 to regulate cytokine expression in melanoma remains unclear. 

We show that the loss of p53 alters two cytokine-dependent phenotypes, chemotaxis and growth factor signaling. Knockdown of p53 in melanoma can reduce melanoma growth [40], which is contrary to its conventional tumor-suppressive role. Our study suggests one potential mechanism for this surprising observation by showing that wildtype p53 can modulate the expression of cytokines, including inflammatory and immune regulators in melanoma cells. These results dovetail with expanding literature on the non-DNA damage roles of p53 in cancer. For some cancers, the benefits of p53 loss in tolerance of mutation and higher proliferative rates may be offset by the loss of non-canonical p53-dependent pathways.

## Figures and Tables

**Figure 1 cells-11-00405-f001:**
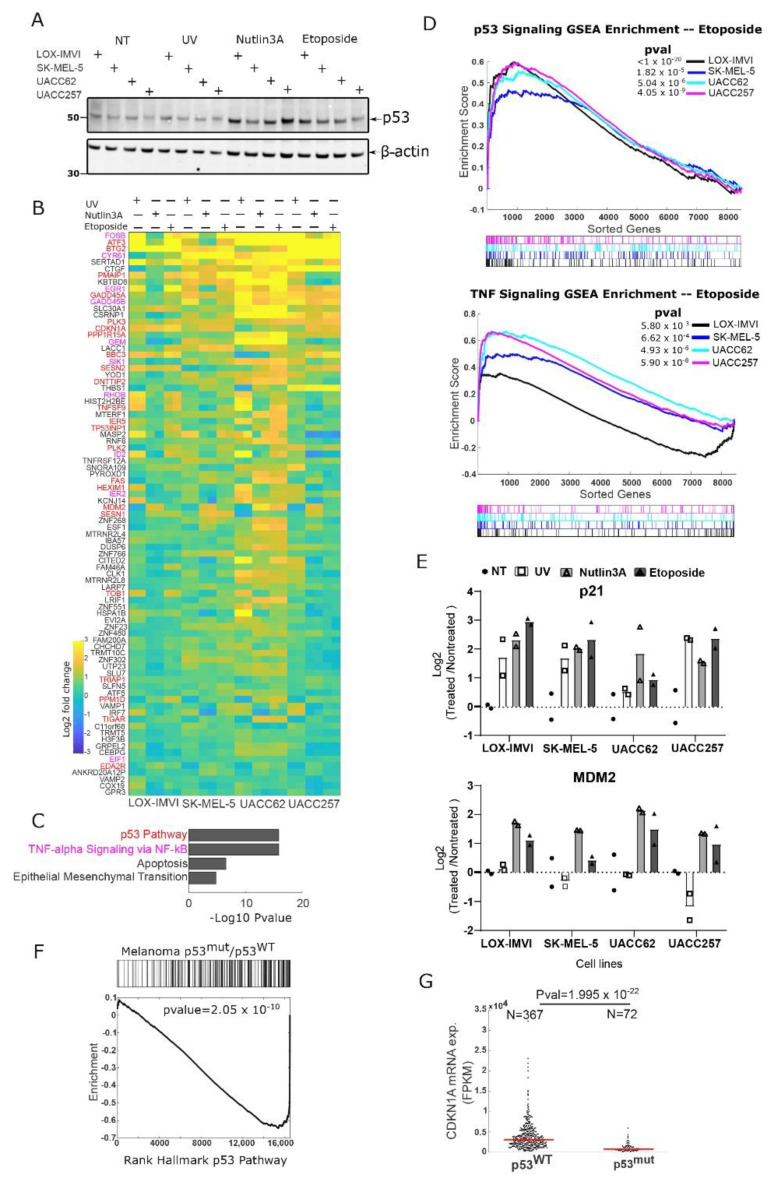
Melanoma cell lines show active p53 signaling on stress. (**A**) Four human melanoma cell lines were not treated (NT) or treated with UV (10 J/m^2^), etoposide (25 µM), or Nutlin-3A (5 µM) for three hours. Western for p53 with β-actin as a loading control. (**B**) Heatmap showing changes in gene expression for each cell line after 3 h of UV, Nutlin-3A, or Etoposide treatment (as in A). Genes whose expression changed in each cell line by at least 1.5-fold in one or more conditions are shown, sorted by average fold change (*n* = 2 for each cell line/treatment). Gene symbol color indicates p53 pathway (red) or NF-kB pathway (purple). (**C**) Gene set enrichment of genes shown in Figure 1B using Hallmark gene sets. (**D**) Gene set enrichment analysis of each cell line for the p53 and TNF signaling pathways for etoposide treatment. (**E**) qPCR data showing the activation of p53 target p21 and MDM2 in the indicated conditions; *n* = 2. (**F**) Gene set enrichment analysis of Gene expression from human melanoma tumors (TCGA samples) comparing tumors with mutant p53 to those with wild type p53. (**G**) mRNA expression of p53 target gene CDKN1A (p21) in human melanoma samples stratified for p53 wild type and p53 mutant alleles; data are mRNAseq collected by The Cancer Genome Atlas (TCGA); units are in fragment per kilobase of transcript per million reads (FPKM).

**Figure 2 cells-11-00405-f002:**
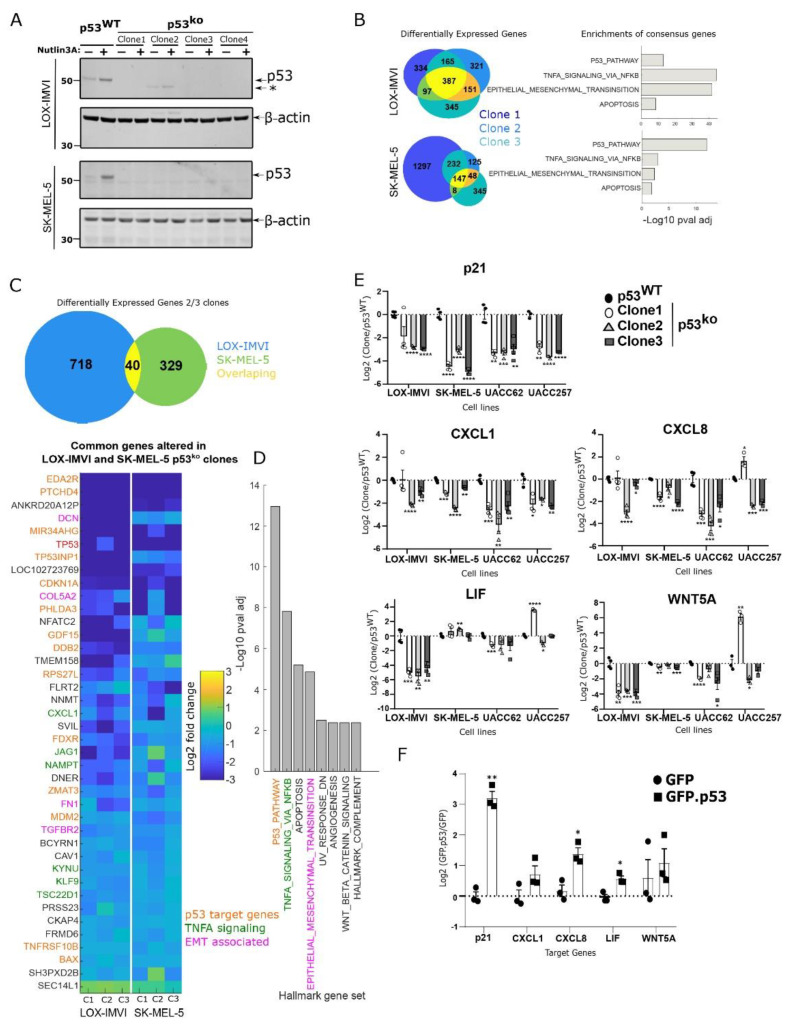
Loss of p53 in melanoma cells results in pleiotropic effects. (**A**) Validation of p53 knockout clones from two cell lines by Western blot. (**B**) Comparison of differentially expressed genes from three p53 knockout clones from LOX-IMVI and SK-MEL-5, enrichment analysis using hallmark gene sets of the overlapping genes from each cell line. (**C**) Differentially expressed genes from each cell line in at least 2/3 p53 knockout clones were identified; expression of each overlapping gene relative to wild type is shown in a heatmap (*n* = 2 for each sample). (**D**) Gene set analysis of overlapping gene set from (**C**). (**E**) qPCR validation of changes in TNF/NF-kB signaling, focusing on a set of key cytokine targets in p53 knockout cells from four melanoma cell lines, *n* = 4, except UACC257 where *n* = 3 (mean ± S.E.M.). (**F**) Expression of wild type p53 in UACC62 p53^ko^ increases expression of p53 target gene p21 and also of cytokine genes identified in (**D**), *n* = 3 (mean ± S.E.M.). Individual biological samples in (**E**,**F**) are represented by a square or circle. Asterisk (*) in (**A**) denotes the lower molecular weight band in one of the LOX-IMVI clones. Significance was determined by *t*-test in compared to wild type in (**E**) or to GFP in (**F**) group, and observed significance level is denoted by an asterisk (* < 0.05; ** < 0.005; *** < 0.0005; **** < 0.0001).

**Figure 3 cells-11-00405-f003:**
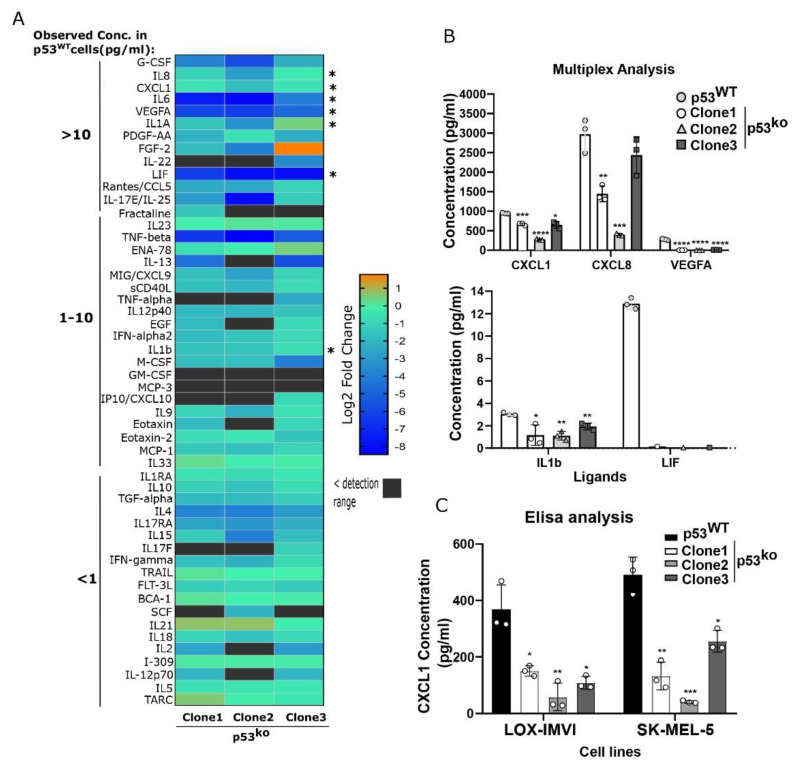
Reduced cytokine secretion in p53 knockout melanoma lines. (**A**) Comparative analysis of ligands abundance in supernatant from p53^ko^ and p53^WT^ LOX-IMVI cells by multiplex cytokine array. Differential expression of secreted ligand for each knockout clone compared to wild type is shown in the heatmap (as in log2-fold change). The observed ligand concentration (pg/mL) in p53^WT^ cells are depicted on the right side by numbers >10, 1–10, and <1. Cytokines analyzed by qPCR (in Figure 2E and Figure 4E) are indicated by an asterisk on the left side. (**B**) Expression of a subset of the ligands in wild-type and knockout clones by multiplex analysis is demonstrated. *n* = 3 (mean ± S.D), except for LIF, in which two biological samples had shown the value below the detection limit. (**C**) Validation of CXCL1 expression in p53^ko^ LOX-IMVI and SK-MEL-5 cells by Elisa analysis; *n* = 3 (mean ± S.D). Significance was determined by *t*-test in comparison to the wild-type group, and the observed level of significance is shown by asterisk (* < 0.05; ** < 0.005; *** < 0.0005; **** < 0.0001).

**Figure 4 cells-11-00405-f004:**
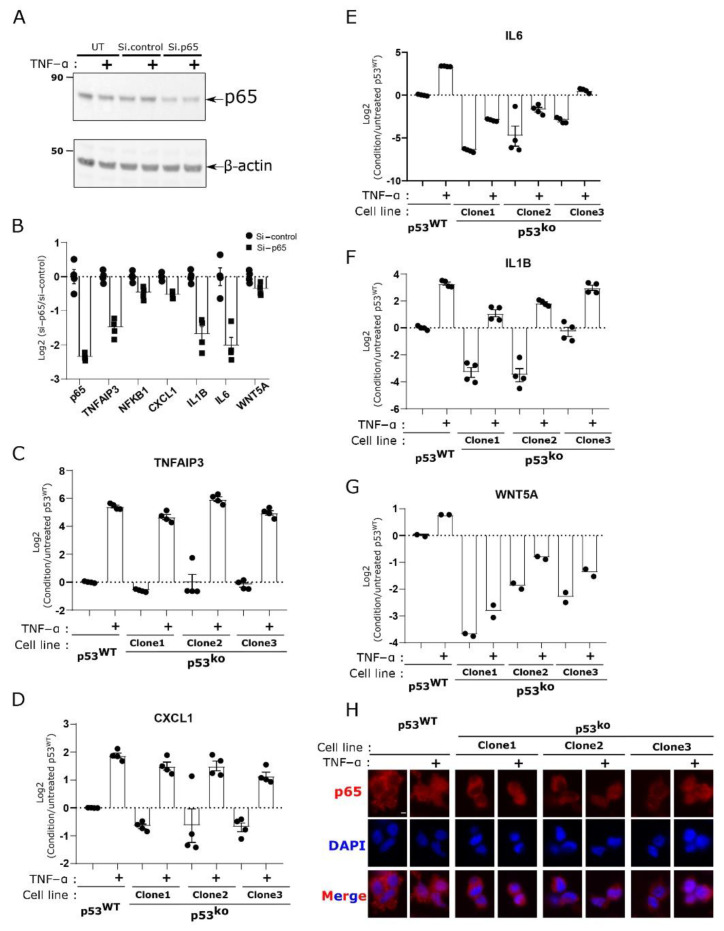
NF-kB activation by TNF-alpha and regulation of cytokine expression does not depend on p53. (**A**) si-RNA-mediated p65 knockdown validated by Western blot analysis. (**B**) qPCR analysis of NF-kB target genes in p65 knockdown LOX-IMVI cells. (**C**–**G**) p53^WT^ and p53^ko^ cells were stimulated with 10 ng/mL TNF-alpha for 2 h, followed by qPCR analysis, focusing on specific sets of NF-kB target genes, TNFAIP3 (**C**), CXCL1 (**D**), IL6 (**E**), IL1B (**F**), and WNT5A (**G**). (**H**) Analysis of p65 expression in p53^ko^ and p53^WT^ LOX-IMVI cells stimulated with 10 ng/mL TNF-α for 15 min, using immunofluorescence. The bar graph in (**B**–**G**) is representing the mean ± S.E.M. of four (**B**–**F**) or two (**G**) biological repeats. Each circle or square dot is representing an individual biological sample. Scale bar in (**H**) is equal to 5 µm.

**Figure 5 cells-11-00405-f005:**
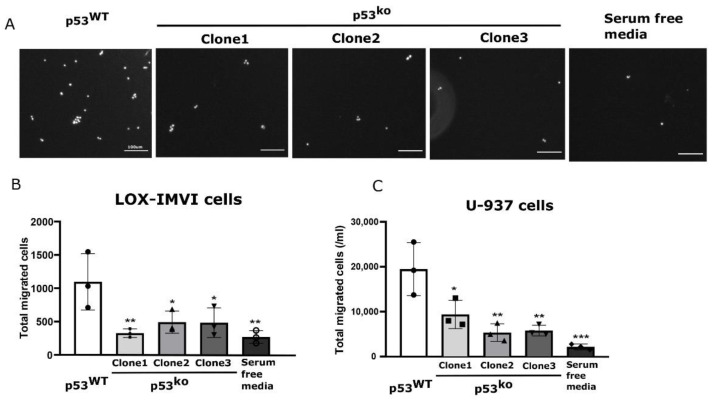
Supernatant from p53 knockout melanoma shows reduced chemoattractant properties. (**A**) A representative transwell image of migrated LOX-IMVI cells in response to p53^WT^ vs. p53^ko^ supernatant. (**B**,**C**) The total migrated LOX-IMVI (**B**) and U-937 (**C**) cells through transwell in response to supernatant is shown by bar graph. *n* = 3, mean ± S.D. for (**B**,**C**); significance assessed by one-way ANOVA relative to the p53^WT^ group. The observed significance level is shown by asterisk (* < 0.05; ** < 0.005; *** < 0.0005). Scale bar in (**A**) is equal to 100 µm.

**Figure 6 cells-11-00405-f006:**
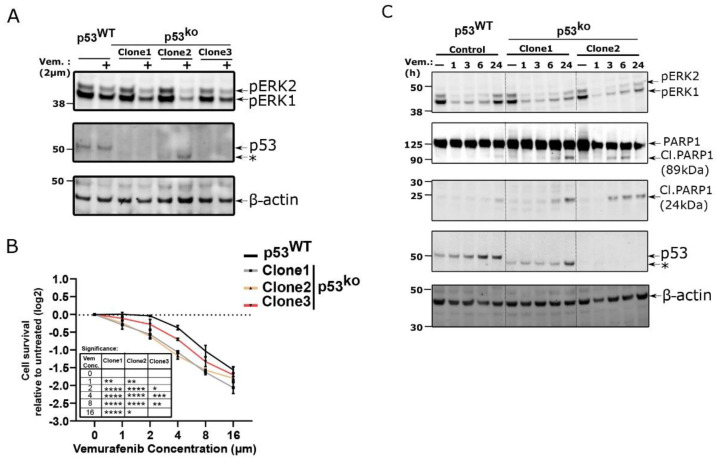
p53 loss in LOX-IMVI cells increases sensitivity towards vemurafenib. (**A**) Western blot for pERK abundance in LOX-IMVI cells exposed to vemurafenib (2 µM, 24 h). (**B**) Cell viability analysis of p53^WT^ and p53^ko^ LOX-IMVI cells after exposure to the indicated dose of vemurafenib for 48 h by CellTiter-Glo. Change in cellular viability (in log2-fold) relative to untreated cells is shown by line graph (*n* = 3, mean ± S.D.). Significance was assessed by two-way ANOVA relative to p53^WT^ group, and the significance level is shown in the inserted table by asterisk (* < 0.05; ** < 0.005; *** < 0.0005; **** < 0.0001). (**C**) p53^WT^ and p53^ko^ LOX-IMVI cell treated with 2 µM vemurafenib for indicated time points and pERK, PARP1 full length and cleavage product abundance measured by immunoblot. Asterisk (*) in (**A**) and (**C**), denotes the lower molecular weight band (as demonstrated in Figure 2A) in one of the LOX-IMVI clones.

## Data Availability

RNAseq data is deposited at GEO (GSE182570,). Other relevant data is included within the manuscript, source data, and Appendix A.

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
