# Peer review of "p53 Promotes Cytokine Expression in Melanoma to Regulate Drug Resistance and Migration"

_cells, 2022, doi:10.3390/cells11030405_

Round 1

Reviewer 1 Report

In the manuscript, the authors found that melanoma cell lines harbor wild-type p53, which can lead to drug resistance and affect metastatic potential. The authors showed that cytokine expression by NFkB signaling was also involved in this process. I think the experiments were well-performed, therefore I have only minor points that should be addressed by the authors. Specific comments are as follows.

Major points:

  1. The authors claim that p53 promotes cytokine expression in melanoma to regulate drug resistance. However, the authors used only the BRAF inhibitor vemurafenib in the study. Treatment with BRAF inhibitors may be effective in metastatic melanoma with BRAF V600 mutations, but disease progression may occur due to resistance to treatment. Therefore, the effects on other types of anticancer drugs should also be investigated.

  1. Overall, western blot bands are not clean. For example, p53 in Figure 6C does not clearly separate WT from KO. This needs to be improved.

Minor points:

  1. The authors should uniform the terminology they use (e.g. NFKB and NF-kb).
  2. Figure 5A: The difference between the images is not clear and needs to be improved.
  3. Figure 6B: It is necessary to evaluate the results using statistical methods.
  4. In western blotting, the location of the molecular weight marker should be indicated.

Reviewer 2 Report

The manuscript entitled “p53 promotes cytokine expression in melanoma to regulate drug resistance and migration” by Pinakin Pandya et al. reports interesting data on the effect of p53-KO melanoma cells lines. The study demonstrates several relevant effects at both DNA and proteins levels.

A few minor issues should be addressed in my opinion:

  • Misspelling “shows” at line 343;
  • Justify the use of the hypergeometric test reported at line 350;
  • Please explain better Figure 3A in the text and in the figure legend; for instance what “>10”, “1-10” and “<1” mean, as referred to the log2foldchange color code. Such relation sounds not clear to me;
  • Please double check sentence at line 410-411;
  • Sentence at line 474 appears an over-statement, regarding the occurrence of apoptosis, which should be shown.

Reviewer 3 Report

Comments and suggestions:

The manuscript cells-1467920 from Pandya et al aims to study the impact of knocking down the transcription factor in melanoma cells, suggesting that in these solid tumors, the presence and persistence of p53 wild type expression can be associated with cytokine expression, in particular key inflammatory cytokines to regulate drug resistance and chemotaxis.

 Overall, I have found that the manuscript was well conceived, with a good experimental design but results presentation should be greatly improved, with more resolution images in figures and with probably a distinct layout of figure composition. In my opinion the results obtained by the authors can be presented with more consistency and objectivity to illustrate the conclusions.

 For this I suggest the authors to consider some prominent changes in the manuscript that I will describe into more detail as follows:

I suggest the authors to reference the first sentence of the introduction. Then, starting with the results presentation, in figure 1, plots D to F need to be enhanced, bigger, with more clear text in a size that any person could read and with good resolution. The same applies for plots found in figure 2E-2F. Still relative the majority of the plots presented by the authors, the statistic treatment is not uniform represented; sometimes the authors use ** to show the significance of the values, whereas other occasions include the p values in the plots. This is not necessary if the uniform statistic treatment is represented in the form of *.

Additionally, in all western-blotting results, the underline IB: sentence of the antibody used can be removed since it is indicated which protein is detected by the antibody. Also, here some discrepancy between nomenclature used in the same blot should be corrected, i.e. ActB and b-actin and in reality, it is not ActB that should be described but instead b-actin.

In figure 2C the maps are with bad resolution and the numbers are not perceptible.

In much of the plot analysis, the y axis is indicated as log2 of something. Here, it should be changed for instance to concentration of a certain cytokine, with the respective units.

I have found hard to see the effect of p53 loss in reduced cell migration since with the representative images shown in figure 5A it is impossible believe in their quantification made in figure 5B. There are no cells in the images in figure 5A that can be used to quantify. If these are the representative images for migrated cell quantification, I cannot understand how the authors were able to quantify total cell migration. This way, I do not agree with the authors in taking conclusion concerning reduced cell migration by loss of p53 based on the results shown. This section must be improved with more clear results and other migration assays that can corroborate the transwell migration.

Throughout the manuscript it should be checked some discrepancies of language, the use of distinct terms to illustrate the same (NFKB, NF-kb in conclusion mainly).

In line 342 it should be indicated as follows: overlapping genes altered in the p53 knockout cells.

In line 349 the p should be written in italic and omit value.

I think it is not necessary to describe in line 368 the reason of using the UACC62 line because it is easier to transduce.

Supplementary figure only has one panel. For this I question the existence of A since there are no more panels for the figure.

Please consider revising in bullet statement what’s already known about this topic; the first bullet point is a copy of the background of the abstract and the second bullet point should be addressed to the already known about mucins in skin, since the authors choose to study specifically MUCL1 in melanocytes. Does not make sense to talk here about pharmaceutical potential targets for melanoma treatment since the manuscript is far from addressing MUCL1 as a novel anti-melanoma target. Also, please consider revising the bullet statement of the translational message, since its true from the results obtained to state the importance of the study to skin dysfunction related with melanogenesis but not with metastasis formation in melanoma.

Round 2

Reviewer 3 Report

Comments to the authors:

Some minor corrections I believe still persist in the present form of the manuscript. Here are my suggestions that have mainly to do the statistical analysis and presentation throughout the manuscript, that in my opinion is not properly addressed.

Please find the points as follows:

1) Figure1G- please describe in the Y-axis of the plot what FPKM stands for and which type of expression? It is not possible to understand from the legend of the figure, neither through the text of section 3.1 of the manuscript how this plot data was generated.

 2) In the legend of the figure 2, where is written "Significance determined by t-test in compared to wild-type (in e) or to GFP (in f) group and observed significance level is denoted by asterisk."; some of the plots in e) and f) have one, two or three for significance. It should be written in the legend the significance of each one, since it is unlikely that in all plots * (0.033); **(0.002); ***(<0.001)). Usually, what is commonly used for significance is to adopt one asterisk for values <0.05, two asterisks for values <0.005 and three for values <0.0005; relating the number of zeros until the number 5 to the number of asterisks.  In the case of the numbers obtained by the authors, 0.002 and 0.001 denote for the same number of asterisks. It is justifiable to include the significance values in the legend and not being in material and methods. Please consider modifications accordingly.

 3) Figure 3C- the same comment applies here for the description of observed the level of significance shown by asterisk; being three distinct asterisks. Are the values determined the same for plots on figure 2? I suggest adopting the description in the respective legend of the meaning of each asterisk.

 4) In figure 4H the resolution of the immunofluorescent images is below the reasonable 300dpi. Here the images are too blur. Please correct the insertion of the images in the field of the figure with the correct resolution: maybe when composing the figure with such small images, is compressing them. Use the raw images and be careful in composing the figure.

5) In figure 5, again which is the value of the significance asterisks for the plots B and C?

6) In legend of figure 6, the significance was assessed by Two-way Anova relative to p53WT group. A distinct test relative the ones used in previous plots. The asterisks here have the same values? Insert please in the legend what is the corresponding value of *, ** and *** of the table inserted.
